# Ultrafast 3D nanofabrication via digital holography

Wenqi Ouyang[1], Xiayi Xu[1,2], Wanping Lu[1], Ni Zhao [3], Fei Han[1,2] ✉ & Shih-Chi Chen [1,2,4] ✉

There has been a compelling demand of fabricating high-resolution complex three-dimensional (3D) structures in nanotechnology. While two-photon lithography (TPL) largely satisfies the need since its introduction, its low writing speed and high cost make it impractical for many large-scale applications. We report a digital holography-based TPL platform that realizes parallel printing with up to 2000 individually programmable laser foci to fabricate complex 3D structures with 90 nm resolution. This effectively improves the fabrication rate to 2,000,000 voxels/sec. The promising result is enabled by the polymerization kinetics under a low-repetition-rate regenerative laser amplifier, where the smallest features are defined via a single laser pulse at 1 kHz. We have fabricated large-scale metastructures and optical devices of up to centimeter-scale to validate the predicted writing speed, resolution, and cost. The results confirm our method provides an effective solution for scaling up TPL for applications beyond laboratory prototyping.

3D printing, which revolutionizes the engineering design and fabrication process[1–5], is one of the most important inventions in the 20th century. As the most effective 3D printing technique at the nanoscale, TPL has been used to create a variety of 3D nanostructures for applications in photonics[6,7], robotics[8,9], machine design[10,11], biomimetic materials[12–14], and metamaterials[15–17] etc. Conventionally, TPL creates structures of arbitrary shapes by serially scanning a focused femtosecond (fs) laser beam in liquid photo-resin. However, such approach is usually too slow for large-scale or practical applications, rendering TPL an exotic and expensive laboratory tool to produce microscale prototypes.

To improve the fabrication rate, a multi-focus scanning strategy that splits a laser beam into multiple points for parallel printing has been developed. Representative examples include multi-beam inteference[18], micro-lens array[19], and diffractive optical element (DOE)[20]. However, as the generated laser foci cannot be individually controlled, these methods are typically limited to writing periodic structures. While the use of programmable beam shapers allows the generation and dynamic control of individual laser foci[21,22], the previously reported methods are either limited by the low device pattern rate or limited number of commandable foci (< 10) due to insufficient laser power[22]. This is because the initiation of two-photon polymerization (TPP)[23,24] requires high laser intensity (~TW/cm²); and using multiple foci directly increases the required laser power. A typical fs laser oscillator, i.e., the most widely used light source in TPL, can hardly provide the peak power to support over 50 laser foci[22]. Recently, a projection-based TPL approach was developed based on a depth-resolved fs light sheet[25]; however, the layer-by-layer fabrication process prevents it from creating large-scale overhanging structures without supporting structures.

A fs regenerative laser amplifier, which has a peak power on the scale of ~10 GW, may well address the problem of insufficient power. Yet, it has rarely been used in TPL, owing to its low repetition rate (1 – 10 kHz vs. 80 – 100 MHz in an oscillator), where previous studies show this could lead to elevated polymerization threshold, reduced dynamic range[24,26–28], and different kinetics during the writing process. Although some of the influences have yet to be explored and understood, it is generally

[1]Department of Mechanical and Automation Engineering, The Chinese University of Hong Kong, Shatin, Hong Kong. [2]Hong Kong Centre for Cerebro-Cardiovascular Health Engineering, Hong Kong Science Park, Shatin, Hong Kong. [3]Department of Electronic Engineering, The Chinese University of Hong Kong, Shatin, Hong Kong. [4]Centre for Perceptual and Interactive Intelligence, Hong Kong Science Park, Shatin, N.T., Hong Kong. ✉e-mail: feihan@cuhk.edu.hk; scchen@mae.cuhk.edu.hk

believed substantial modifications in optical configurations, photoresists, and printing parameters are needed to apply such laser sources for TPL. In addition, whether similar structural integrity and sub-microscale resolution can be achieved with the laser amplifier during multi-focus operation has yet to be investigated.

We present a multi-focus TPL platform for ultrafast 3D nano-printing based on digital holography. The system employs a 1 kHz fs regenerative laser amplifier to support parallel writing with up to 2000 hologram-generated laser foci. All laser foci can be individually controlled in terms of amplitude, phase, and location. A photoresist is custom-developed to work with the ultrahigh-peak-power light source. The kinetic factors that determine the quality of printing, e.g., the diffusion and solidification processes, have been investigated to optimize the system. Based on these results, the TPL system achieves a volume printing speed of up to 54.0 mm³/h, and a resolution of 90 and 141 nm in lateral and axial directions, respectively. The random-access scanning capability enabled by the digital holography makes our system particularly efficient in printing low material filling ratio structures, e.g., 1 – 12%. We have fabricated large-scale mechanical metastructures, arrays of magnetic micro-gears, 2D diffractive surfaces, and other complex 3D nanostructures to demonstrate the system performance.

## Results

### Optical setup and synthesis of binary hologram

The optical configuration of the multi-focus TPL system is presented in Fig. 1a, where the light source is a Ti:sapphire fs laser amplifier with an 800-nm central wavelength, 1 kHz repetition rate, 100 fs pulse width, and 4 W average power. A digital micro-mirror device (DMD) displays the designed holograms, which generate the laser foci on the Fourier plane after L3; a spatial filter is placed there to block the unused diffraction orders. The holograms are synthesized based on the weighted Gerchberg-Saxton (WGS) algorithm[22,29], which contain the designed laser foci with pre-determined amplitude, phase, and location with high accuracy. By sequentially displaying the synthesized holograms on the DMD (synchronized with the laser pulses), the laser foci can perform high-speed random-access scanning and 3D fabrication (i.e.,

2000 foci × 1000 Hz = 2,000,000 voxels/sec). To eliminate the angular dispersion introduced by the DMD (owing to its small pixel sizes), which would broaden the laser pulses, a reflective blazed grating (600 lines/mm) and a 4-f system (L1 and L2) are placed before the DMD to pre-compensate the dispersion (Supplementary Fig. 2). Lastly, the laser foci on the DMD Fourier plane are projected to the photoresist under the objective lens via a second 4-f system (L4 and L5).

The multi-focus printing process and results are presented in Fig. 1b-d. A fluorine-doped tin oxide (FTO) glass substrate is mounted on a hexapod six-axis positioner under the objective lens. A high numerical aperture (NA) oil immersion objective is used for dip-in fabrication in the liquid photoresist with matching refractive index. Next, the laser foci perform parallel scanning along the designed trajectories to fabricate the nanostructures. Lastly, solid structures are formed after developing the printed substrate with propylene glycol monomethyl ether acetate (PGMEA) and isopropanol (IPA). To print structures of sizes larger than the DMD scanner's work volume (i.e., 299 × 554 × 760 μm³ in the x, y and z directions), the sample positioner can move around in six axes to stitch the structures on demand.

### Design of the photoresist

With the ultrahigh peak power, the regenerative laser amplifier can support up to 2000 laser foci for parallel printing. Yet this presents a drastically different printing condition comparing with conventional TPL processes. As known from previously reported TPP models[28,30], the focus intensity of our laser system (i.e., 3.3 – 22.7 TW/cm²) is sufficiently high to directly initiate multiphoton ionization of the photo-initiators (Supplementary Figs. 3 and 4, and Supplementary Note 1 and 2). In order to suppress the ionization process while promoting two-photon absorption (2PA), we have selected initiators with a large 2PA cross-section. Specifically, a molecule with bis-donor structure and symmetrically substituted conjugated chains was used (Fig. 2a)[25,31]. The 2PA cross-section of this molecule at 800 nm is estimated to be ~800 GM, which is nearly two orders larger than commercial photo-initiators, such as Irgacure 819 and 754. To increase the mechanical strength of the resin, 4,4'-(4,4'-isopropylidenediphenoxy)-bis-(phthalic anhydride) (BPADA, 68 wt%) is mixed with pentaerythritol triacrylate (PETA, 32 wt%) to form the base of monomers. Next, the initiator molecules

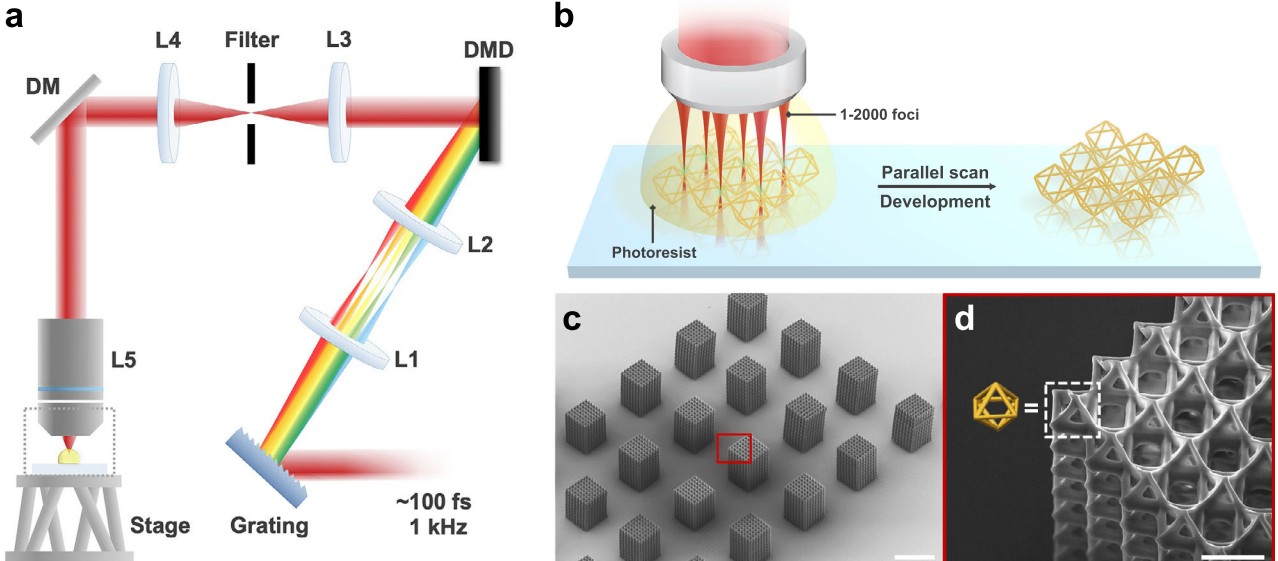

**Fig. 1 | Configuration and operation principle of the multi-focus TPL system. a** Optical setup of the TPL fabrication system. DM: dichroic mirror; L1-L4: lenses ($f_{L1}$, $f_{L2}$, $f_{L3}$, $f_{L4}$ = 225, 250, 150, and 200 mm, respectively); L5: objective lens; (see Supplementary Fig. 1 for more details.); **b** printing process of a 3 × 3 octahedral truss structure via multi-focus scanning; **c** SEM images of an octahedral truss structure printed via 64 foci; and **d** zoom-in view of the red box in **c**, where the yellow structure shows a unit cell. Scale bars are 100 μm for (**c**), and 10 μm for (**d**).

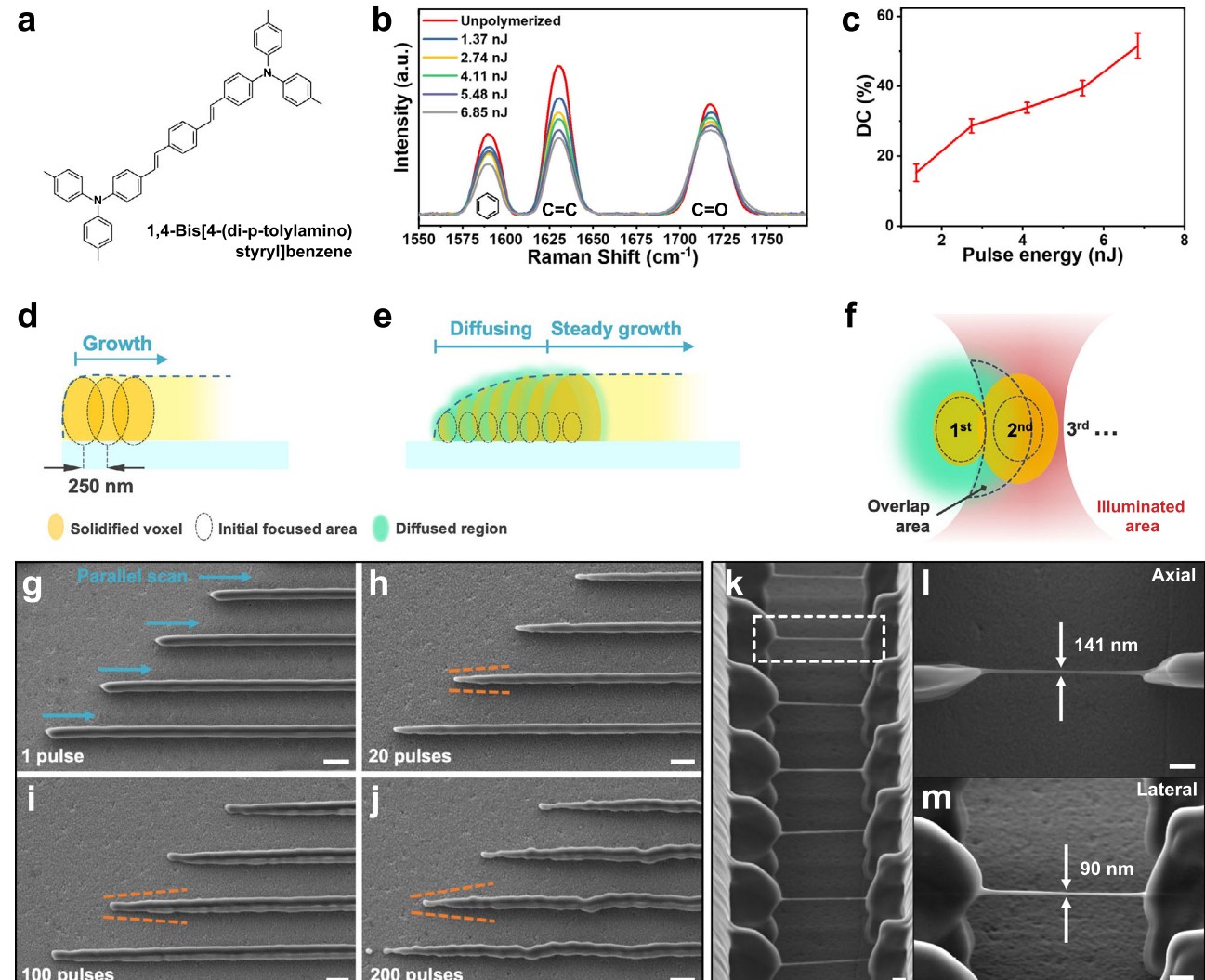

**Fig. 2 | Optimization of the TPL system under the ultrahigh-peak-power laser.**
**a** Molecular structure of the photo-initiator. **b** Raman spectra of the photoresist polymerized with different single pulse energies; **c** DC plotted as a function of (single) pulse energy; **d**, **e** schematics illustrating the polymerization kinetics of fabricating the nanowires "without" and "with" the influence of diffusion respectively; **f** zoom-in view of the polymerization condition of the initial voxels in **e**, showing how polymerization of the later voxel is influenced by the diffusion effect. **g**–**j** SEM images of nanowires fabricated by 1, 20, 100 and 200 pulses (per voxel), respectively; **k** an array of free-standing nanowires printed by 10 foci; and **l**, **m** side view and isometric view of the dashed box in **k** that show a 141 and 90 nm axial and lateral resolution, respectively. Scale bars are 1 μm.

are dissolved in the monomer mixture. After testing different concentrations, the optimal resin consists of 0.4 wt% of the initiator, and 70 ppm of 4-hydroxyanisole as inhibitor. The ultraviolet-visible absorption spectrum of the photoresist was measured and presented in Supplementary Fig. 5.

The photoresist has achieved a low polymerization threshold of 1.27 nJ under single-pulse exposure and a wide dynamic range of 12.46 based on the regenerative laser amplifier. With such low polymerization threshold, the photoresist can achieve sufficient degree of cross-linking (DC) with relatively low single-pulse energy. The DC of the photoresist with different pulse energies was estimated via Raman spectroscopy. In the experiment, three characteristic peaks at 1589, 1630, and 1717 cm$^{-1}$ were obtained in the liquid photoresist (Fig. 2b), indicating the presence of benzyl group, carbon double bond, and carbonyl group, respectively. When the pulse energy increased from 1.37 to 6.85 nJ, the peak intensity of the double bond significantly decreased due to polymerization. Accordingly, the corresponding DC is calculated[32] to be increased from 16.2 to 48.5% (Fig. 2c). The result confirmed that our fabrication platform can perform TPL with a single-

pulse energy of less than 10 nJ, enabling a new printing strategy to be discussed in the next section.

## Single-pulse TPP fabrication model for low-repetition-rate lasers

In general, to write solid structures in liquid photoresists, the laser dose in each voxel should be sufficiently high for continuous gelation and solidification of the photoresist. The size of each voxel is determined by the diffusion radius of the starting radicals and partially reacted monomers. Thus, when the exposure duration of a voxel as well as the time interval between two successive exposures are controlled within the diffusion threshold (e.g., ~20 ms for PETA to diffuse 150 nm[28]), unwanted voxel expansion beyond the laser focal volume can be minimized. Fig. 2d-f graphically illustrate the printing process without and with the influence of diffusion. The diffusion effects (Fig. 2e) are typically not observed in conventional TPL processes, which are based on high-repetition-rate laser oscillators (i.e., 80 – 100 MHz). The photoresists are continuously polymerized by low-power laser pulses (e.g., ~10,000 pulses to fully cure a voxel). Thus, the diffusion of the partially reacted species cannot be observed as each voxel's exposure time is

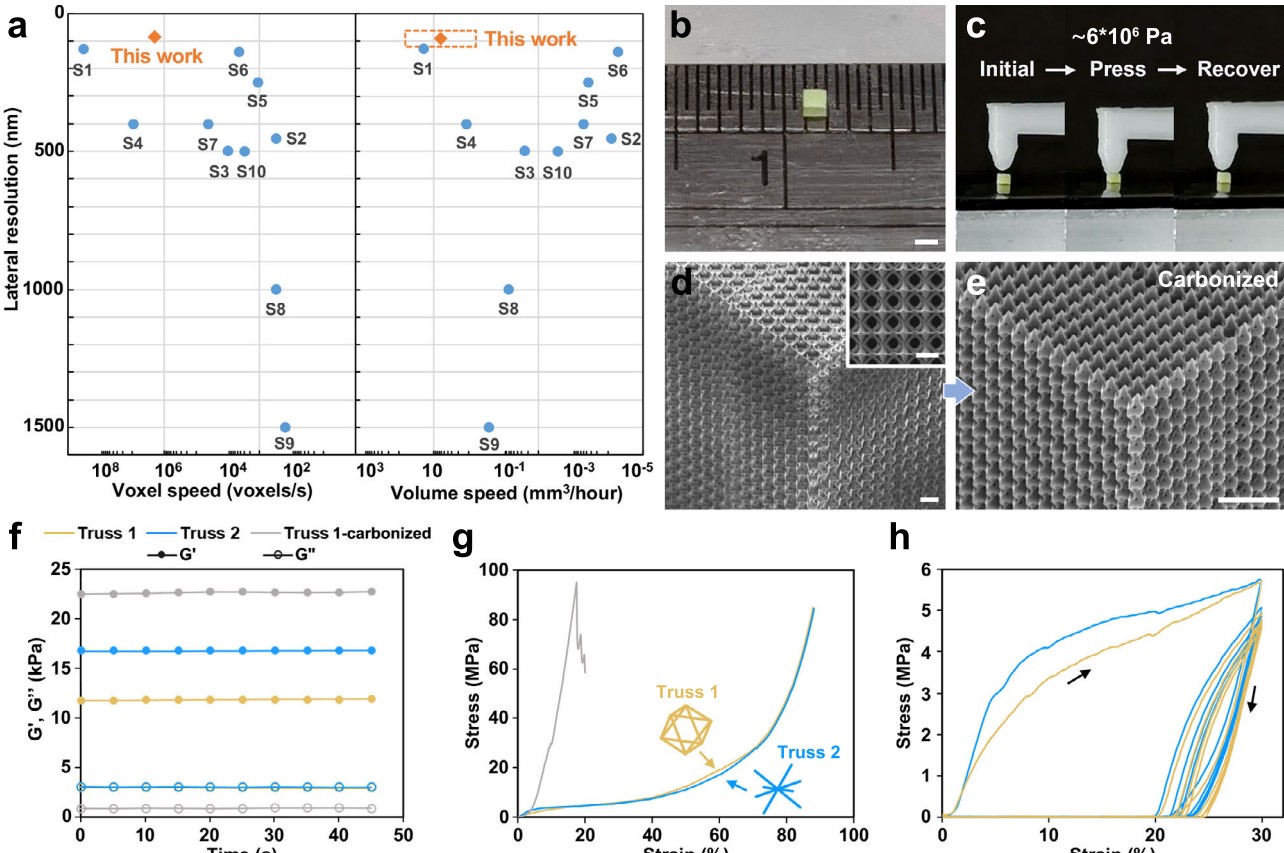

**Fig. 3 | Characterization of the fabrication rate by printing large-scale 3D metastructures. a** Comparison of printing rate and resolution of our method with other competing TPL solutions. Referenced works include: S1[25], S2[21], S3[22], S4[20], S5[19], S6[44], S7[45], S8[46], S9[47], S10[48]. The data for the chart are presented in Supplementary Table 3. **b** Photograph of a cubic metastructure with a volume of 1.17 mm³ and sub-units of octahedral trusses. **c** Compressing and recovering processes of the cubic metastructure. **d** SEM images showing the structural details of the cube, where the inset shows the top view of the structure; **e** SEM image of the cube after carbonization. **f** Storage modulus (G') and loss modulus (G") curves of the two octahedral trusses (Truss 1 and 2), and the carbonized Truss 1 recorded from the time sweep tests. Young's modulus of the two trusses were measured to be 35 and 80 MPa, respectively. **g** Compression stress-strain curves of the three trusses in **f**, where carbonized metastructures have a reduced fracture strain of ~17%. **h** Stress-strain curves of uniaxial cyclic compression of the two trusses. The plot shows that the 1st, 5th, 10th, 15th, and 20th loading-unloading cycles. Scale bars are 1 mm for (**b**), and 10 μm for (**d**) and **e**.

always less than a few milliseconds (Fig. 2d). In comparison, for the 1 kHz laser, when each voxel's exposure time is longer than diffusion threshold (≥ 20 pulses), the diffusion effect becomes increasingly prominent (Fig. 2e). As illustrated in Fig. 2f, the partially reacted species (green) may diffuse into the vicinity of the second exposure's focal region (red) and become polymerized, which increases the voxel size and eventually forms the tapered structure before it reaches a steady state. The diffusion limit (i.e., the steady-state linewidth) is determined by the laser intensity around the focal spot: Although the partially reacted polymers can diffuse to the exposure region of the next voxel, the light intensity and size of the focal spot is constant throughout the printing process. Thus, the voxel size cannot expand further when the partially reacted polymers diffuse outside of the region with sufficient laser intensity because polymers with low DC cannot survive the post-treatment processes. To verify our hypothesis, we simulated the intensity distribution at the focal spot with the printing parameters in Fig. 2i. The results are presented in Supplementary Fig. 6, which suggest that the diffusion limit is ~ 700 nm in the lateral direction, and beyond which the light intensity is insufficient (< 0.1 TW/cm²) to polymerize the diffused species. This agrees with the experimental results in Fig. 2i, where the linewidth at the steady state is ~720 nm.

To minimize the influence of diffusion, we adopt the single-pulse exposure strategy (i.e., each voxel is polymerized via a single laser pulse), which simultaneously achieves the best resolution and scanning rate. It is worthwhile to note that this strategy is opposite to the

conventional TPL process that demands a combination of low laser power and long exposure time to create high-resolution features. In other words, our system breaks the trade-offs between resolution and rate. Fig. 2g-j present an array of parallel nanowires printed using 1-, 20-, 100-, and 200-pulse exposure processes with identical total laser doses. All structures were fabricated using 20 laser foci and a step size (voxel distance) of 250 nm. As the laser pulses increase from 1 to 200, one may observe the starting end of the nanowire forms a tapered shape with increasing surface roughness. The length of the tapered section also increases with the number of laser pulses. As the laser pulses continue to accumulate, minor deviations of light can multiply during the exposure, leading to roughened surfaces and distorted structures, i.e., at 200 pulses, the nanowire became distorted, as predicted by our model (Fig. 2j). To examine whether high-pulse energy during the single-pulse printing process would induce avalanche ionization, we fabricated an array of nanowires, each with different pulse energies (from 3 to 18 nJ), and measured their horizontal and axial widths. The results are presented in Supplementary Fig. 7, where an abrupt direction change in the trendlines was observed at the pulse energy of ~10 nJ. This suggests polymerization induced by avalanche ionization occurred at high average laser intensity (> 10 nJ or 25.9 TW/cm²)[33,34]. As this energy is much higher than those used in our printing experiments (i.e., 3 – 7 nJ), it is reasonable to believe that multiphoton absorption is the dominant mechanism of polymerization in our single-pulse printing process.

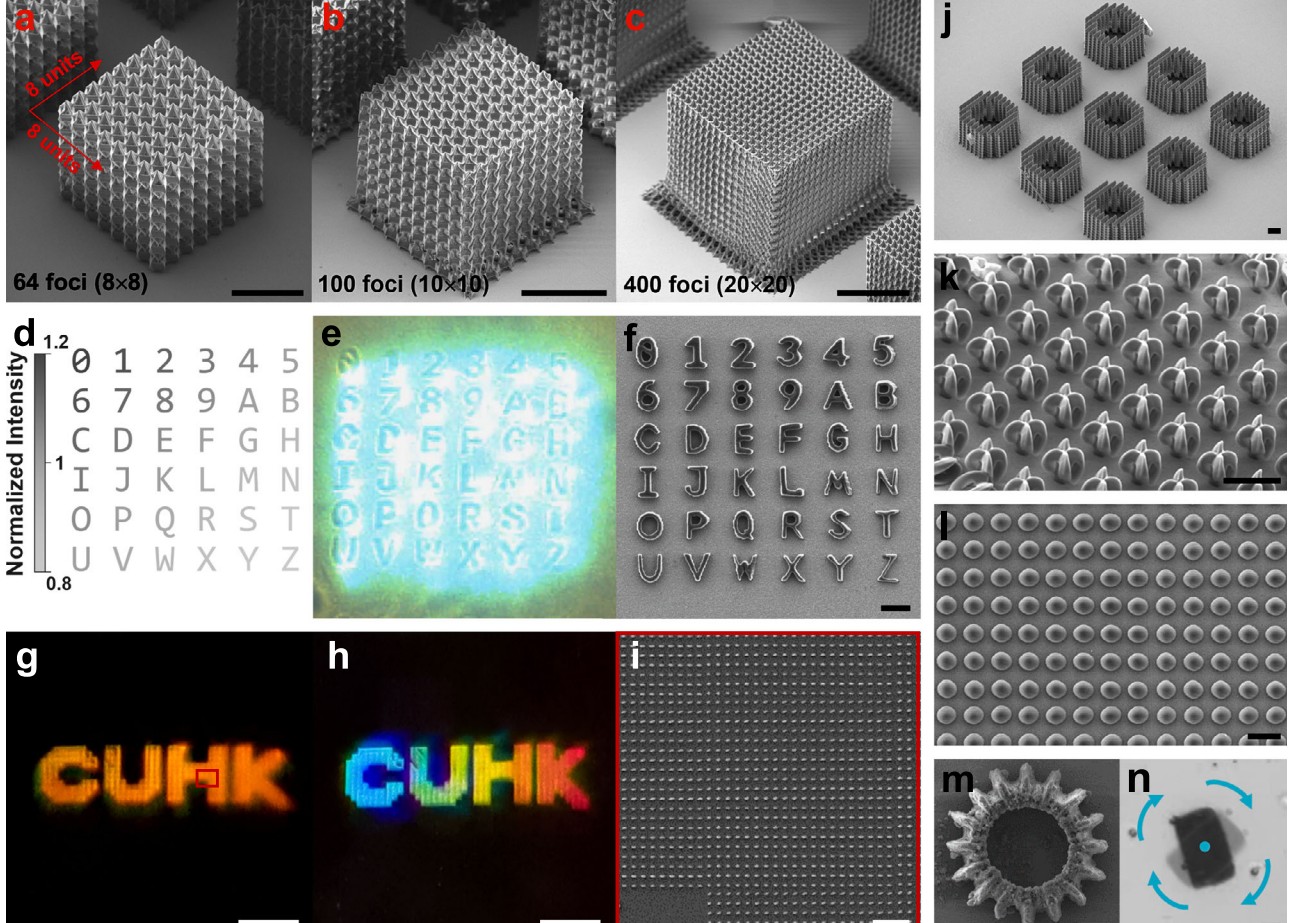

**Fig. 4 | Fabrication of complex 3D nanostructures using 36 − 400 foci. a−c** Meta-structures printed by 64, 100, and 400 foci, respectively (with an average pulse energy per focus of 5 nJ). **d** Designed alphabets and numbers to be printed with different laser intensities. **e** Image of in situ 36-focus fabrication. **f** SEM image of the fabrication results of (**e**). **g**, **h** Optical images of a "CUHK" pattern consisting of nano-dot arrays with a period of 4 μm; the mono- or rainbow-color effects are induced by the reflection / diffraction of light. **i** Zoom-in view of the red box in **g**; **j** Array of annular structures formed by woodpile sub-units. **k** Array of self-supported egg-beater micro-structures. **l** Array of micro-lenses. **m** Magnetic micro-gear, and **n** time-lapse image of the micro-gear driven by a magnetic field. Scale bars are 50 μm for (**a–c**); 10 μm for (**f**) and **i−m**; and 1 cm for (**g**) and **h**.

## Optimization of multi-focus fabrication

With the new findings and understandings, we apply the single-pulse exposure strategy to simultaneously achieve the best resolution and scanning rate in the following experiments. We also vary the number of laser foci to optimize the fabrication rate and resolution. First, to test the resolution, we fabricated an array of hanging nanowires using 10 laser foci, which reached a lateral and axial resolution of 90 and 141 nm, respectively, with high uniformity (Fig. 2e-g). To the best of our knowledge, these features are thinner than any previously reported results using parallel[18–22] or projection-based fabrication approaches[25].

In our preliminary studies of multi-focus writing, we observed that over-polymerization can occasionally occur when 20 or more laser foci are simultaneously employed, resulting in expanded linewidths or even unwanted bulk solidification (Supplementary Fig. 8b: bulk solidification above a patterned structure). This may be attributed to (1) the close proximity of the scanning foci and the diffusion of the nearby reaction species, which together cause laser doses to accumulate in a specific region, and (2) nonuniform laser power distribution in the diffraction envelope that overlaps with the DMD work volume (Supplementary Fig. 8a). Similar phenomenon has been observed and studied by Arnoux, C. et al.[35], which agrees with our observation. Further experimental investigation shows that the bulk solidification effect can be entirely eliminated, achieving high-quality printing results, for up to 2000 laser foci when the smallest distance between any two scanning foci is larger than 3 μm (Supplementary Fig. 9). In parallel, the diffusion effect can be largely suppressed by the single-pulse exposure strategy (Fig. 2a-d); and the laser intensity distribution can be improved to 99% via the WGS algorithm. In fact, our laser system can potentially support parallel printing with over 4000 foci (Supplementary Fig. 10). Yet this high printing rate comes at the expense of slightly compromised focus quality, laser intensity distribution, and the possibility of over-polymerization owing to the high proximity of laser foci. These negative effects may be alleviated by using a low-magnification objective lens, which can increase the DMD work volume (at the expense of slightly increasing the minimum step size).

## Characterization of the fabrication rate

The fabrication rate of TPL is often described in two formats, i.e., printed volume per unit time (e.g., mm³/hour) or printed voxels per unit time (e.g., voxels/s). The two formats have been used interchangeably in the past because although they can be quite different in describing the fabrication of a solid structure versus a low volume-filling ratio structure, the numbers are the same for a conventional point-scanning TPL system, i.e., the laser focus has to scan through the entire build volume regardless of the structure porosity due to the limitation of inertia-based scanners, e.g., galvanometric mirror. For our DMD scanner, which performs multi-focus random-access 3D scanning, the fabrication rate for low volume-filling ratio structures can

be substantially increased because the laser foci do not need to scan over regions without solid structures (Supplementary Fig. 11). In addition, the fabrication time remains constant for any designed fabrication trajectories. As shown in Fig. 3a, the fabrication rate of most parallel TPL methods are below 0.1 mm³/hour or $10^4$ voxels/sec. In comparison, when printing hollow structures with a filling ratio of 1 – 12% (i.e., the ratio of most lattice materials[36,37]), our system can reach a rate of 4.5 – 54.0 mm³/hour when 2000 foci are employed (Supplementary Fig. 12), which is among the best reported TPL volumetric processing rates. At sub-200 nm resolution, our system has achieved a voxel printing rate of up to $2 \times 10^6$ voxels/s, which is better than state-of-the-art commercial solutions by three orders of magnitude.

Notably, although two methods in Fig. 3a (S2 and S1), i.e., DOE-based multi-focus system[20] and projection-based TPL[25], show superior throughputs, our system still has clear advantages in resolution, energy efficiency, and fabricating complex and low volume-filling ratio structures. For example, due to its fixed scanning points, the DOE-based system can only achieve the promised rate when printing periodic patterns; and the projection-based TPL method has to compromise its fabrication rate when printing grayscale structures. In contrast, our method can achieve precise grayscale control (accuracy > 99%, Supplementary Fig. 13) via the single-pulse exposure process as the amplitude, phase and position of each laser focus is encoded in the design holograms. Further optimization of the hologram can realize advanced functions, including intensity error compensation and multi-focus beam shaping (Supplementary Fig. 14). Regarding energy efficiency, owing to the ultrahigh peak power of the laser amplifier, our system requires much lower average power (20 – 400 mW for 100 – 2000 foci) then the aforementioned two systems: The DOE-based system[20] requires an operating power of ~4 W, and the projection-based TPL[25] requires ~1.5 W to achieve the best fabrication rate. Overall, our method has achieved the best resolution, energy efficiency, and flexibility among all high-throughput TPL methods.

### Fabrication and characterization of large-scale metastructures

High-throughput fabrication of functional metastructures consisting of hundreds to thousands of units with sub-micrometer scale features has been one of the greatest challenges in nanotechnology. Due to the limited fabrication efficiency, previous studies typically fabricate such structures with only hundreds of meta-units[11,38,39], or with size-scale gaps of several orders of magnitude, which prevent them from showing the predicted bulk properties[40]. To address this issue, we fabricated a $1.08 \times 1.08 \times 1$ mm³ metastructure that contains $6.82 \times 10^5$ units of octahedral trusses[41] (Truss 1, Fig. 3b and d) using 100 foci. Each meta-unit has a side length of 7.9 µm, and a linewidth of 700 nm. As shown in Fig. 3c, the cube can be compressed for ~30%, and fully recovered after the load is removed.

We next studied the mechanical properties of the metastructure via a rotating rheometer. A second metastructure consisting of a different type of octahedral truss[42] with the same porosity (Truss 2) was fabricated and tested as a control group. First, we measured the dynamic modulus of the two metastructures via oscillatory time-sweep tests, where Truss 2 shows slightly higher stiffness (Fig. 3f). In the stress-strain tests, the two metastructures show almost identical responses (Fig. 3g), where no signs of fractures were observed at 85% strain (Supplementary Fig. 15). Next, we tested the mechanical responses of the two metastructures via cyclic compression tests from 0 to 30% strain (Fig. 3h), where plastic deformation was observed for both structures in the first cycle. After that, similar loading cycles were observed in the next 5 – 20 testing loops. These experiments, which show 20 – 30% mechanical resilience, have proved the predicted high compressibility characteristics of mechanical metamaterials[36,37]. The similar mechanical responses of the two metastructures indicate the fabrication process is reproducible with reliable performance.

Lastly, Truss 1 was carbonized at 900 °C in nitrogen, which reduced the scale of the metastructure by ~73% to form a carbonized structure (Fig. 3e). The isotropic shrinking result indirectly suggests that the internal structures of Truss 1 was uniform; and the photoresist was appropriately polymerized. The carbonized structure achieves much higher stiffness ($G' = 22.7$ kPa) and Young's modulus (497.3 MPa), which is close to the modulus of solid polymers (Young's modulus = 520 MPa, Supplementary Fig. 16)[43]. These results show the multi-focus TPL platform to be a powerful tool for fabricating and studying macroscale metastructures.

### Fabrication of complex nanostructures and nanodevices

First, we studied the relation between the part quality and the number of laser foci. Three cubes consisting of octahedral trusses and supporting bases were fabricated using 64, 100, and 400 laser foci, respectively (Fig. 4a-c and Supplementary Movie 1). From the results, we can confirm the quality of the printed structures were not compromised by the use of increasing laser foci.

Next, we demonstrate grayscale writing and independent focus control via fabricating a 2D array of numbers and alphabets (Fig. 4d) using 36 laser foci, where each focus carries different laser powers to achieve 11 different grayscale levels. As shown in Fig. 4e and Supplementary Movie 2, each number or alphabet was fabricated by a single laser focus; and the total fabrication time is reduced to 150 ms. The SEM image (Fig. 4f) confirms the quality of the fabrication results. To demonstrate large-scale fabrication, a $1 \times 4$ cm² "CUHK" pattern consisting of nano-dot arrays was printed (Fig. 4g-i) to function as a 2D diffractive surface. By adjusting the incident angle of a coherent white light, mono- or rainbow-color can be generated. The uniform color distribution indirectly suggests accurate size and periods of the nano-dot array. We have also fabricated other 3D structures to demonstrate the precision and reproducibility of our system, including an array of annular structures formed by woodpile sub-units, self-supported egg-beater structures, and a micro-lens arrays (Fig. 4j-l, respectively).

The multi-focus TPL system can fabricate functional micro-machines with high throughput. As a proof-of-concept demonstration, we prepared a magnetic photoresist by mixing polyacrylic acid functionalized $Fe_3O_4$ nanoparticles (8 nm, 3.0 wt%) with our photoresist. Notably, the resulting photoresist has significantly increased writing threshold (8.6×), which is a known drawback of such composite-based photoresists. Although this is a problem for conventional TPL systems that greatly reduces the throughput (e.g., prolonged exposure time and reduced voxel distance[10]), our multi-focus TPL system can effortlessly work with the photoresist due to the high peak power laser source. Fig. 4m and Supplementary Fig. 17 present an array of magnetic micro-gears, fabricated via 32 foci with a voxel distance of 500 nm. The total fabrication time for the array was 49 seconds (or 290 ms for a single micro-gear). Upon removal from the substrate, the micro-gear was remotely controlled by a 10 mT magnetic field in an aqueous environment to demonstrate complex robotic motions including directional moving, rotation (5 Hz), and flipping (1 Hz) (Fig. 4n, and Supplementary Movie 3 and 4).

## Discussions

The multi-focus TPL platform has achieved the best-reported resolution, the most flexible scanning strategy, the most accurate grayscale control, and one of the highest fabrication rates among all parallel TPL solutions. This result is enabled by the unique photopolymerization kinetics, fs laser amplifier, custom-designed photoresist of high dynamic range, and holography-based DMD scanner. With the improved rate, precision and reduced cost, the multi-focus TPL platform may find important applications in producing functional nano-/micro-structures at large scale, thereby creating new opportunities in the fields of photonics, flexible electronics, material sciences, mechanics, biomedicine, and micro-robotics.

## Methods

### Materials

PETA (technical grade), BPADA (97%), 4-hydroxyanisole (MEHQ, 99%), and isopropanol (IPA, 99%) were purchased from Sigma Aldrich. $Fe_3O_4$ nanoparticles were purchased from US Research Nanomaterials Inc. PGMEA (99%) was purchased from Dieckmann. Photoresist IP-Dip™ was purchased from Nanoscribe. All chemicals/photoresists were used without further purification.

### Printing steps

First, an FTO-coated glass substrate (surface conductivity of 20 ohm) was pre-cleaned to remove dusts or possible oxidations. The substrate was sequentially sonicated for 20, 10, and 8 min in glass cleaning fluid, DI water, and IPA, respectively. Next, the substrate was dried in an 80 °C-oven for two hours. For printing, the cleaned substrate was placed on the sample stage, and a drop of photoresist was dispensed on it. Next, the laser focus array was generated in the photoresists by the holograms to fabricate the designed structures. After printing, the patterned substrate was post-processed in PGMEA for 15 min and then in IPA for 10 min. After development, the patterned substrate was dried in open air at room temperature. For fragile structures (e.g., free-standing wires of less than 300 nm-widths or metastructures that tend to distort by surface tension), a freeze dryer was used to maintain their shapes: The patterned substrate was immersed in DI water and frozen at −40 °C for 12 hours, which was then freeze-dried in a FreeZone 2.5 Liter −84C Benchtop Freeze Dryer from Labconco for another 12 hours.

### Characterizations

SEM images were obtained using a JEOL JSM-7800F field-emission scanning electron microscope at an acceleration voltage of 5 or 10 kV on a tilt stage. Before SEM imaging, the samples were coated with a layer of Pt using an Edwards Sputter Coater to enhance its conductivity. The optical images were collected via a COSSIM CMY-310 optical microscope. Raman spectrums were obtained via an inVia™ confocal Raman microscope system from Renishaw under a 532 nm laser.

### Preparation of photoresist

The photo-initiator molecules were synthesized according to reported methods[25,31]. 0.4 wt% of the initiator molecules were dissolved in a monomer mixture of PETA (32 wt%) and BPADA (68 wt%) under vigorous sonication for 5 hours. 70 ppm of MEHQ was then added to the mixture, followed by another 1 hour of sonication.

### Preparation of magnetic photoresist

The functional photoresist is a ferrofluid of nanoparticles and photoresist. $Fe_3O_4$ nanoparticles (3.0 wt%) were mixed with our custom-designed photoresist using a planetary micro mill (QM-3SP04 from Nanjing Instrument) for 30 min to ensure the mixture is uniform. The magnetic actuation of the micro-gears was performed in an aqueous environment containing polyvinylpyrrolidone (6.0 wt%, Aladdin), and a poly-(methyl methacrylate) chamber as the platform. The programmable magnetic field was generated by Helmholtz coils, and magnetic fields of 10 mT were applied to control the micro-gears.

### Testing the mechanical performances

The mechanical properties of the metastructures were tested on a rotating rheometer (Malvern KINEXUS Lab + ). After affixing the sample on the testing plate, oscillatory time-sweep measurements (strain 1%, frequency 1 Hz) were conducted to obtain the dynamic moduli (G′ and G″) of the sample. For static compression tests, the sample was compressed at a fixed strain rate of $0.1\,s^{-1}$ to the target strain levels.

Cyclic loading-unloading hysteresis experiments with 20 repeating cycles were performed at a strain rate of $0.1\,s^{-1}$.

### Carbonization of the metastructures

The polymeric metamaterials were carbonized at 900 °C for 1 hour in a tube furnace (model OTF-1200X) filled with nitrogen with a ramp rate of $5\,°C\cdot min^{-1}$ to the target temperature, and were then naturally cooled to the room temperature. The process resulted in a ∼ 73% linear shrinkage of their initial dimensions.

## Data availability

The data that support the findings of this study will be available from the corresponding author upon request.

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

## Acknowledgements

We thank Dr. Fengtong Ji and Prof. Li Zhang for performing tests on the magnetic micro-gears; Prof. Xijiang Han and Mr. Ce Liang for obtaining Raman spectrums; Mr. Songyun Gu for useful suggestions in establishing the optical platform. This project was partially supported by the Research Grants Council, General Research Fund (14209421, 14203020); Innovation and Technology Commission, Innovation Technology Fund (ITS/178/20FP); Science, Technology and Innovation Commission of Shenzhen Municipality (STIC) (SGDX2020110309500100), and InnoHK Centre projects funded by the Innovation and Technology Commission (A-CUHK-16-5-14) and (COCHE-1.5).

## Author contributions

S.C.C., F.H., and W.O. conceived the study. W.O. designed and built the optical system and performed all of the fabrication experiments. F.H. designed the photoresists and studied the polymerization kinetics. X.X. tested the mechanical performance of the printed metastructures. W.L. tested the parameters of the photoresists on a Ti:Sapphire laser oscillator. N.Z. assisted in designing the experiments. F.H. and W.O. prepared the figures and schemes together. F.H. and W.O. prepared the original manuscript with revisions from S.C.C.

## Competing interests

The authors declare no competing interests.
