## [Peer Review File · Nature Communications]

Ultrafast 3D Nanofabrication via Digital HolographyReviewer #1 (Remarks to the Author):

See attached file

Reviewer #1 Attachment on the following page

What are the noteworthy results?

Noteworthy results that are shown is that they have been able to take a time-consuming nanofabrication process and improve upon it without sacrificing resolution. This improvement has allowed them to fabricate large meta-structures and optics via this nanofabrication method at a much higher throughput than before (roughly three orders of magnitude faster). This brings the TPL process a step closer towards being an effective solution for applications outside of prototyping in the laboratory.

Will the work be of significance to the field and related fields? How does it compare to the established literature? If the work is not original, please provide relevant references.

The work shown holds significance in its field and potentially others. As previously mentioned, current methods of TPL are time consuming and not viable outside of laboratory prototypes. Not only have they shown a method in how to scale the TPL process to increase its speed, but they also use a single-pulse strategy rather than the convention process of long exposure with low power, the new strategy yielding higher resolution and scan rates without having to trade off. It will potentially have significance in other fields if the TPL process can be scaled further without sacrificing its high resolution. If that can be achieved, it may be possible to see this technology help develop other fields of nanotechnology such as energy, biomedicine, etc.

This is unique compared to established literature due to how it approaches TPL, and results achieved. While there is literature out there that seeks to improve the scanning speed, they take different approaches than shown here.

Does the work support the conclusions and claims, or is additional evidence needed?

The work does support the conclusions and claims made via the information, figures, and methodology provided. Additional figures showing the physical setup would be beneficial but can also be deemed unnecessary. However, it is mentioned that structures larger than the work volume can be printed by the sample positioned moving in six axes. Can this introduce errors in the part? How do you ensure repeatability in the stage movement?

Are there any flaws in the data analysis, interpretation, and conclusions? Do these prohibit publication or require revision?

None noted.

Is the methodology sound? Does the work meet the expected standards in your field?

The methodology seems sound. It provides detailed steps to reproduce the work and results provided with equipment that can measure targeted resolution and mechanical properties. They also provided and used their own standard operating procedures (SOP) that should yield similar results if they were to make/test a new sample.

Is there enough detail provided in the methods for the work to be reproduced?

The information provided in the methodology section is enough for the most part to reproduce the work presented. They provide materials used, steps done prior and after printing, and the equipment used to verify both resolution and mechanical performance.

Questions:

How are we defining “low filling ratio structures”? I’m assuming that they’re structures printed that have a specific infill or are otherwise hollow.

What would have to be done differently to enable 4000 foci and address the possible issues with possible compromised focus quality, laser intensity distribution, and possible over-polymerization? Would this require significant changes to the current setup to achieve the same quality of print?

Reviewer #2 (Remarks to the Author):

The reported results are reaching new level of resolution and writing speed capabilities. it is also shown that very different photo-materials can be used.

Low repetition rate and single voxel per pulse irradiation allows to purify discussion about the mechanism of polymerization. The higher the intensity, the more efficient is multiphoton absorption (hence two photon as the first nonlinear process), which competes with avalanche ionisation dominant at lower intensities. Evolution of diameter of polymerised line of its axial extent vs pulse intensity (or fluence or pulse energy) would clearly reveal what is the polymerisation mechanism
<https://doi.org/10.1515/nanoph-2020-0551>.

From the mechanical tests it would be useful to extract the bulk material properties since the volume fraction is known (Appl. Phys. Lett. 91, 241904 (2007);
<https://doi.org/10.1063/1.2822825>)

Figure 2 is discussed in terms of diffusion of reactive species required for polymerisation. Equally relevant is thermal diffusion which is driven by strong gradient of absorbed energy deposition.

Please show in the supplement absorbance spectra of the used photo-materials/resists.

Reviewer #4 (Remarks to the Author):

In this manuscript, the authors utilize the digital holography-based two-photon lithography (TPL) platform and custom-developed photoresist to realize the controllable multi-focus parallel printing, which breaks the trade-off between resolution and rate. Through investigating the kinetic factors, high quality printing results are achieved. With the significantly improved writing speed, resolution and grayscale controllability, micro/nanostructures with various functions have been fabricated, which will expand the application in photonics, mechanic, micro-robotics and so on. This work is generally well organized and written. However, the current manuscript lacks detailed analysis. Therefore, a revision is required before it can be published on Nature Communications.

1. As the structure fabrication is based on the multi-focus parallel scanning strategy, how is the connectivity of the adjacent sites between two units? Is it just a simple superposition? If so, the repeated scanning at the common point or on the common axis will make the size inconsistent with the designed one, which decreases the printing precision and accuracy. In addition, the authors mention that "To print structures of sizes larger than the DMD scanner's work volume (i.e., $299 \times 554 \times 760 \mu\text{m}^3$ in the x, y and z directions), the sample positioner can move around in six axes to stitch the structures on demand", how the connection of the interface between two adjacent structures is, which is important and will influence the structural integrity.

2. Please provide more detailed description in the Methods section.

(1) The function of the fluorine-doped tin oxide glass substrate.

(2) How about the post-processed process? Is the printed structure only rinsed and soaked with propylene glycol monomethyl ether acetate and isopropanol? The unreacted photoresist filling in the gaps of the structure with low volume-filling is difficult to be completely removed by rinsing. In addition, the detailed cleaning process should be provided.

(3) The supporting bases are clearly visible in Fig. 4a-c, which are attached to the printed structures. The authors should illustrate how the supports are removed from the structures, and add this as the post-treatment step in the Methods section. Furthermore, whether it will influence the printing precision and structural integrity after the supporting structure being removed?

3. The mechanism of polymerization kinetics with the influence of diffusion is not explained clearly. As shown in Fig. 2e, how is the boundary between the diffusing region

and steady growth region defined? The triggered free radicals gradually accumulate during the continuous scanning process, so it is difficult to understand why the steady growth state exists and the printed structure with nonuniform width should be obtained theoretically as the printing proceeds. Is the subsequent steady growth caused by the depletion of locally diffused free radicals by the oxygen inhibitor effect? In addition, as the authors mentioned, the nanowires printed through single-pulse exposure strategy should have the best printing quality, why the nanowires fabricated by 20 pulses (Fig. 2h) are thinner than the 1 pulse (Fig. 2g).

4. Some details need to be verified and revised.

(1) Reorder the images in the supplementary information according to the logic in the main manuscript.

(2) The description in the Note of Supplementary Figure 13 is "The results in Supplementary Fig. 12i-j show that carbonized metastructures have substantially reduced plastic range", however, there are no panel i and j in Supplementary Fig. 12, please verify and correct.

(3) In the dynamic modulus test, as shown in the Fig. 3f, the Truss 1 has higher stiffness than Truss 2. However, the description in the manuscript is "First, we measured the dynamic modulus of the two metastructures via oscillatory time-sweep tests, where Truss 2 shows slightly higher stiffness", the conclusion does not agree with experimental result.

Response to Reviewers' Comments

Reviewer 1:

Comments to the Author:

What are the noteworthy results?

Noteworthy results that are shown is that they have been able to take a time-consuming nanofabrication process and improve upon it without sacrificing resolution. This improvement has allowed them to fabricate large meta-structures and optics via this nanofabrication method at a much higher throughput than before (roughly three orders of magnitude faster). This brings the TPL process a step closer towards being an effective solution for applications outside of prototyping in the laboratory.

Will the work be of significance to the field and related fields? How does it compare to the established literature? If the work is not original, please provide relevant references.

The work shown holds significance in its field and potentially others. As previously mentioned, current methods of TPL are time consuming and not viable outside of laboratory prototypes. Not only have they shown a method in how to scale the TPL process to increase its speed, but they also use a single-pulse strategy rather than the convention process of long exposure with low power, the new strategy yielding higher resolution and scan rates without having to trade off. It will potentially have significance in other fields if the TPL process can be scaled further without sacrificing its high resolution. If that can be achieved, it may be possible to see this technology help develop other fields of nanotechnology such as energy, biomedicine, etc. This is unique compared to established literature due to how it approaches TPL, and results achieved. While there is literature out there that seeks to improve the scanning speed, they take different approaches than shown here.

Response:

We thank the reviewer for the positive comments and for recognizing our innovation in developing the TPL platform and the single-pulse printing strategy.

Comments to the Author:

Does the work support the conclusions and claims, or is additional evidence needed?

The work does support the conclusions and claims made via the information, figures, and methodology provided. Additional figures showing the physical setup would be beneficial but can also be deemed unnecessary. However, it is mentioned that structures larger than the work volume can be printed by the sample positioned moving in six axes. Can this introduce errors in the part? How do you ensure repeatability in the stage movement?

Response:

Thank you for your comments. When printing structures larger than the working volume of the optical scanner, stitching is needed and performed by the precision sample stage. It is well known that stitching will introduce errors, including errors from the sample stage as well as slight structure volume change

during photopolymerization (which is unavoidable). Stage errors can be minimized by using a precision stage; in our experiments, a high-precision six-axis positioner (H-811.I2, Physik Instrumente; repeatability = ± 60 nm) was used to minimize the stitching errors. To minimize volume (or density) change after polymerization, a polymer with high mechanical strength was used in our photoresist (i.e., BPADA, 68 wt%). Accordingly, stitching errors in all our experiments have been well controlled and minimized to a semi-indistinguishable level even under a scanning electron microscope, e.g., the large 3D structures shown in Fig. 3b-e and Supplementary Fig. 15. Below, we show a 3D octahedron truss metastructure (Fig. R1) that has an area of 0.9×0.69 mm² area, which again confirms that stitching errors have been controlled and minimized.

Fig. R1. SEM images that show (a) the top surface of a printed octahedron truss metastructure, and its zoom-in view (b); inset shows the structural details.

Comments to the Author:

Are there any flaws in the data analysis, interpretation, and conclusions? Do these prohibit publication or require revision?

None noted.

Is the methodology sound? Does the work meet the expected standards in your field?

The methodology seems sound. It provides detailed steps to reproduce the work and results provided with equipment that can measure targeted resolution and mechanical properties. They also provided and used their own standard operating procedures (SOP) that should yield similar results if they were to make/test a new sample.

Is there enough detail provided in the methods for the work to be reproduced?

The information provided in the methodology section is enough for the most part to reproduce the work presented. They provide materials used, steps done prior and after printing, and the equipment used to verify both resolution and mechanical performance.

Response:

We thank the reviewer for the positive comments about our methodology and the provided standard operating procedures.

Comments to the Author:

Questions: How are we defining “low filling ratio structures”? I’m assuming that they’re structures printed that have a specific infill or are otherwise hollow.

Response:

Thank you for your question. The “low filling ratio structures” refers to “low material filling ratio structures”, which are hollow or porous structures. Notably, most mechanical metamaterials structures fall under this category, which typically has a material filling ratio (in relative to the hollow part) below 25%. We have revised the description of this term in page 2, line 13 to avoid confusion.

Comments to the Author:

What would have to be done differently to enable 4000 foci and address the possible issues with possible compromised focus quality, laser intensity distribution, and possible over-polymerization? Would this require significant changes to the current setup to achieve the same quality of print?

Response:

Thank you for your question. Indeed, our system can generate 4000 or more laser foci for parallel processing; they were not used in the experiments because of the small work volume ($299 \times 554 \mu\text{m}^2$ in x-y plane) set by the objective lens, i.e., when too many foci appear and work in a small confined space, proximity effect tends to happen (discussed in Supplementary Materials Fig. 8), which can compromise the focus quality and make trajectory planning difficult. When a low magnification objective lens is used, the work volume will be increased, and accordingly, more laser foci can effectively work together. Yet, since our scanning process is discrete, a low magnification lens also means the minimum step size (or resolution) will be increased.

A discussion about the printing with 4000 foci is included on page 6, line 6 in the revised manuscript.

Reviewer 2:

Comments to the Author:

The reported results are reaching new level of resolution and writing speed capabilities. it is also shown that very different photo-materials can be used.

Response:

We thank the reviewer for giving us positive comments and recognizing the advances in printing resolution, speed, and the new photoresists.

Comments to the Author:

Low repetition rate and single voxel per pulse irradiation allows to purify discussion about the mechanism of polymerization. The higher the intensity, the more efficient is multiphoton absorption (hence two photon as the first nonlinear process), which competes with avalanche ionisation dominant at lower intensities. Evolution of diameter of polymerised line of its axial extent vs pulse intensity (or fluence or pulse energy) would clearly reveal what is the polymerisation mechanism <https://doi.org/10.1515/nanoph-2020-0551>.

Response:

Thank you for your comments and suggestion. Indeed, multiphoton absorption may compete with avalanche ionization at high light intensity ($>1 \text{ TW/cm}^2$). To examine whether avalanche ionization plays a role in our printing process. Following your advice, we measured the horizontal and axial width of a set of polymer wires printed by sequentially varying the pulse energy of single-pulse exposures. The results are presented in Fig. R2 below.

Fig. R2. Experimental results that show the evolution of (a) linewidth, and (b) aspect ratio vs. pulse energy in a set of polymer nanowires printed via single pulse exposures, respectively. (Note that without photo-initiators, our resin cannot be directly polymerized by our optical system.) The trendlines, which were obtained via logarithmic function or linear function fitting, in (a) are plotted to clearly indicate the changing of trends.

From Fig. R2, we observed an abrupt direction change in the data in both axial and lateral directions at a pulse energy of ~ 10 nJ, which may be attributed to the high average light intensity (~ 25.9 TW/cm²). This suggests that polymerization driven by avalanche ionization may occur at a pulse energy of 10 nJ or higher (Malinauskas, M. et al. *X-photon laser direct write 3D nanolithography*. 2022, Researchsquare, preprint); and accordingly, polymerization caused by multiphoton absorption may be dominant at a lower pulse energy. As pulse energy above 10 nJ is rarely used in our printing experiments (we typically use 3-7 nJ), it is reasonable to believe that multiphoton (including two-photon) absorption is still the dominant polymerization mechanism in our printing process, which is consistent with our hypothesis in the manuscript.

To provide a more solid discussion about the polymerization mechanism, Fig. R2 and the related discussions are included in Page 4, Line 35 of the revised manuscript. The mentioned paper, and the preprint from Malinauskas, M. et al. that help us to analyze into the process are included as new references (32, 33) in the revised manuscript.

Comments to the Author:

From the mechanical tests it would be useful to extract the bulk material properties since the volume fraction is known (Appl. Phys. Lett. 91, 241904 (2007); <https://doi.org/10.1063/1.2822825>)

Response:

Thank you for your comment. In fact, we already measured the Young's modulus of the bulk polymer material by printing a solid polymer cube (520 MPa, Supplementary Fig. 16) in the original manuscript. This is a more direct measurement than extracting it from the volume fraction of a complex 3D structure. We understand that when the bulk material properties cannot be directly measured, a good way to estimate the material properties is to calculate them from the volume fraction of the printed structures. Thus, we have included the mentioned paper as a new reference (42) in the revised manuscript.

Comments to the Author:

Figure 2 is discussed in terms of diffusion of reactive species required for polymerisation. Equally relevant is thermal diffusion which is driven by strong gradient of absorbed energy deposition.

Response:

Thank you for your comment. Indeed, in the conventional study of TPP kinetics under a femtosecond laser oscillator, thermal effect can promote diffusion to a certain degree because the exposure of thousands of laser pulses per voxel under high repetition rate (typically 10^3 - 10^4 pulses/voxel at 80 - 100 MHz) can cause heat accumulation. In our case, although the pulse energy is much higher, the repetition rate is much lower (1 kHz), i.e., the time interval between pulses (1 ms) is much longer than the heat diffusion time constant (~ 1 ns). As such, thermal diffusion effect is typically not considered a dominant factor in processes performed under a femtosecond laser amplifier system.

Comments to the Author:

Please show in the supplement absorbance spectra of the used photo-materials/resists.

Response:

Thank you for your suggestion. The UV-vis adsorption spectrum of our custom-designed photoresist is now included in Supplementary Fig. 5.

Reviewer 4:

Comments to the Author:

In this manuscript, the authors utilize the digital holography-based two-photon lithography (TPL) platform and custom-developed photoresist to realize the controllable multi-focus parallel printing, which breaks the trade-off between resolution and rate. Through investigating the kinetic factors, high quality printing results are achieved. With the significantly improved writing speed, resolution and grayscale controllability, micro/nanostructures with various functions have been fabricated, which will expand the application in photonics, mechanic, micro-robotics and so on. This work is generally well organized and written. However, the current manuscript lacks detailed analysis. Therefore, a revision is required before it can be published on Nature Communications.

Response:

We thank the reviewer for the positive comments as well as the recognition of our new TPL process and printing results. We have provided detailed discussions, analysis, and additional experiments to address the mentioned issues.

Comments to the Author:

1. As the structure fabrication is based on the multi-focus parallel scanning strategy, how is the connectivity of the adjacent sites between two units? Is it just a simple superposition? If so, the repeated scanning at the common point or on the common axis will make the size inconsistent with the designed one, which decreases the printing precision and accuracy. In addition, the authors mention that “To print structures of sizes larger than the DMD scanner’s work volume (i.e., $299 \times 554 \times 760 \mu\text{m}^3$ in the x, y and z directions), the sample positioner can move around in six axes to stitch the structures on demand”, how the connection of the interface between two adjacent structures is, which is important and will influence the structural integrity.

Response:

Thank you for your questions. Although all generated laser foci can perform random-access scanning, each focus is programmed to scan continuously to ensure the printed voxels will overlap and have good structural strengths. We can design and optimize the fabrication trajectory to avoid multiple exposures of a single point (e.g., a common axis). Notably, over-exposure at the juncture point or common axis is a typical issue in conventional raster-scanning systems, but since the scanning path of each focus can be arbitrarily programmed by the DMD scanner, our method can easily avoid such issues and achieve more uniform exposure, as evidenced by the many fabrication results.

Regarding stitching errors, when printing structures larger than the working volume of the optical scanner, stitching is needed and performed by the precision sample stage. It is well known that stitching will introduce errors, including errors from the sample stage as well as slight structure volume change during photopolymerization (which is unavoidable). Stage errors can be minimized by using a precision stage; in our experiments, a high-precision six-axis positioner (H-811.I2, Physik Instrumente; repeatability = $\pm 60 \text{ nm}$) was used to minimize the stitching errors. To minimize volume (or density)

change after polymerization, a polymer with high mechanical strength was used in our photoresist (i.e., BPADA, 68 wt%). Accordingly, stitching errors in all our experiments have been well controlled and minimized to a semi-indistinguishable level even under a scanning electron microscope, e.g., the large 3D structures shown in Fig. 3b-e and Supplementary Fig. 15. Below, we show a 3D octahedron truss metastructure (Fig. R3) that has an area of $0.9 \times 0.69 \text{ mm}^2$ area, which again confirms that stitching errors have been controlled and minimized.

Fig. R3. (a) SEM image shows the top surface of a printed octahedron trusses metastructure, and its zoom-in view (b); inset shows the structural details.

Comments to the Author:

2. Please provide more detailed description in the Methods section.
(1) The function of the fluorine-doped tin oxide glass substrate.

Response:

Thank you for your comments. Fluorine-doped tin oxide (FTO) glass is widely used as substrates of TPL. On the one hand, it has high thermal tolerance (up to $700 \text{ }^\circ\text{C}$) to endure possible heat effects during the laser scanning process; on the other hand, it can effectively increase the adhesion of polymer species (*Polymers* 2021, 13, 2419), which helps prevent the separation between the printed structures and the substrate during other post treatment steps. (Since FTO substrates are commonly used in TPL works, we did not include additional discussions of it in our Method section.)

Comments to the Author:

- (2) How about the post-processed process? Is the printed structure only rinsed and soaked with propylene glycol monomethyl ether acetate and isopropanol? The unreacted photoresist filling in the gaps of the structure with low volume-filling is difficult to be completely removed by rinsing. In addition, the detailed cleaning process should be provided.

Response:

Thank you for your question and suggestion. As described in the “Methods” section, most of our printed structures are post-treated by sequentially soaking in propylene glycol monomethyl ether acetate and isopropanol for 15 and 10 min, respectively. For complex, fragile, or large-scale 3D structures (e.g., metastructures), an additional washing step (with DI water) followed by freeze-drying is used to remove the residual photoresist, and to maintain the printed shapes. Based on our experience, such treatments can effectively remove the remaining photoresists within the structure, which is evidenced by the clean interior of the fractured carbonized metastructure shown in Supplementary Fig. 15i-j. Note all above mentioned post-processing steps have already been included in the Method section.

Comments to the Author:

(3) The supporting bases are clearly visible in Fig. 4a-c, which are attached to the printed structures. The authors should illustrate how the supports are removed from the structures, and add this as the post-treatment step in the Methods section. Furthermore, whether it will influence the printing precision and structural integrity after the supporting structure being removed?

Response:

Thank you for your question. Generally, supporting bases are not needed in our printing method. In the case of Fig. 4a-c, the supporting bases do not need to be removed as the printed structures were designed to demonstrate the quality and precision of multi-focus parallel printing. To provide a fair comparison among different printing results (especially for the complex and fragile metamaterial structures), the supporting bases were included to minimize the possible structure distortion in the post-treatment processes (e.g., washing and freeze-drying). If not particularly mentioned, all printing experiments were performed without the inclusion of supporting bases.

Comments to the Author:

3. The mechanism of polymerization kinetics with the influence of diffusion is not explained clearly. As shown in Fig. 2e, how is the boundary between the diffusing region and steady growth region defined? The triggered free radicals gradually accumulate during the continuous scanning process, so it is difficult to understand why the steady growth state exists and the printed structure with nonuniform width should be obtained theoretically as the printing proceeds. Is the subsequent steady growth caused by the depletion of locally diffused free radicals by the oxygen inhibitor effect? In addition, as the authors mentioned, the nanowires printed through single-pulse exposure strategy should have the best printing quality, why the nanowires fabricated by 20 pulses (Fig. 2h) are thinner than the 1 pulse (Fig. 2g).

Response:

Thank you for your comments and questions. There are no defined boundaries between the diffusion and the steady growth region. Instead, as discussed in our manuscript, the steady state is gradually reached with the continuous printing of more voxels. For the ease of understanding, we briefly

summarize the mechanism as follows:

The diffusion rate of the polymers in our photoresist stays at a level of ~ 7.5 nm/ms (Baldacchini, Tommaso, ed. *Three-dimensional microfabrication using two-photon polymerization: fundamentals, technology, and applications*. William Andrew, 2015, Chapter 3). Our printing process is discrete and the distance between adjacent voxels is typically set to 250 nm. Hence, when the exposure time of each voxel exceeds ~ 20 ms (i.e., 20 pulses/voxel for the 1 kHz laser), the partially reacted species (radicals and partially crosslinked monomers) from the first voxel would diffuse to the vicinity of the second exposure's focal region and become polymerized, which increases the size of the second voxel. With such effect, voxels printed in the later stage would continuously expand until the accumulation of diffused species reaches a limit, that is, the steady growth state.

On this base, the limit of such diffusion effect (i.e., the final linewidth at the steady growth state) is determined by the laser intensity around the focal spot: Although the partially reacted species could always diffuse to the vicinity region of the next voxel, the laser intensity and the size of the focus spot is constant during the printing process. This means when the diffused species exceeds the area that the laser intensity could further polymerize them, the volume of this voxel cannot expand anymore as polymers with low degree of crosslinking would be washed away in the post-treatment processes; accordingly, the steady growth state is achieved.

To better elaborate the mechanism, we simulated the intensity distribution at the focal spot according to the printing parameters in Fig. 2i as an example (100 pulses/voxel, and pulse energy 4.5 nJ). Figure R4b below presents the intensity distribution of the focal point along the lateral direction, where point A is the polymerization boundary of the 1st pulse; and point B is the final polymerization boundary (i.e., size) of the 1st voxel. As the minimal average intensity to induce TPP is ~ 0.1 TW/cm², the simulation results suggest that the limit of diffusion distance is ~ 700 nm in the lateral direction (i.e., $0.351 \mu\text{m} \times 2 \approx 700$ nm at point C in Fig. R4b, where the light intensity is 0.12 TW/cm²), and beyond which the diffused species can no longer be effectively polymerized. This roughly agrees with the experimental results, where the linewidth at the steady growth state is measured to be ~ 720 nm in the lateral direction in Fig. 2i.

Fig. R4. (a) Simulated laser intensity distribution at the focal spot according to the printing parameters in Fig. 2i; (b) plot showing the focal point intensity distribution at $z = 0$ in the lateral dimension. The simulation was performed according to the scalar-based point-spread function model

in the literature (Journal of Microscopy, 2013, 249, 13-25), where the refractive index was set to 1.52, and NA of the objective lens was 1.3.

Lastly, the linewidths fabricated by 1 and 20 pulses were actually within the error margins (with a difference of < 5%). Note that the scale bars in the two images (Fig. 2g and h) are slightly different, which causes the linewidth in Fig. 2h to look thinner. For your reference, we measured the average widths of the four wires in the two figures (from top to bottom): For Fig. 2g, the measured widths are ~440, 450, 460, and 470 nm; for Fig. 2h, the widths are ~420, 430, 440, and 450 nm. (Note that the generally increasing widths for the bottom lines may be attributed to the slightly changing viewing angle of the SEM.)

The discussions here about how the steady growth state is achieved and the simulation results are included on Page 4, line 14 of the revised manuscript; and in the Supplementary Fig. 6.

Comments to the Author:

4. Some details need to be verified and revised.

(1) Reorder the images in the supplementary information according to the logic in the main manuscript.

Response:

Thank you for your suggestion. We have reordered the images to ensure they appear according to the narrative order in the main text.

Comments to the Author:

(2) The description in the Note of Supplementary Figure 13 is “The results in Supplementary Fig. 12i-j show that carbonized metastructures have substantially reduced plastic range”, however, there are no panel i and j in Supplementary Fig. 12, please verify and correct.

Response:

Thank you for your comment. The Supplementary Fig. 12i-j should be Supplementary Fig. 13i-j instead. We have corrected this error in the revised Supplementary file.

Comments to the Author:

(3) In the dynamic modulus test, as shown in the Fig. 3f, the Truss 1 has higher stiffness than Truss 2. However, the description in the manuscript is “First, we measured the dynamic modulus of the two metastructures via oscillatory time-sweep tests, where Truss 2 shows slightly higher stiffness”, the conclusion does not agree with experimental result.

Response:

Thank you for pointing out our mistake. It is Truss 2 that has a higher stiffness in the dynamic modulus tests; and the color was incorrectly labeled in Fig. 3f. We have corrected this error in the revised manuscript.

Reviewer #1 (Remarks to the Author):

All of my comments have been sufficiently addressed for this publication

Reviewer #2 (Remarks to the Author):

review queries are well answered and additional information is provided to support the answers.

Reviewer #4 (Remarks to the Author):

The authors have carefully revised their manuscript according to the reviewers' comments. The mechanism of how the steady growth state is achieved has been explained in a clearer way. I am pleased to recommend this paper for publication on Nat. Commun. in its current form.

Response to Reviewers' Comments

Reviewer 1:

Comments to the Author:

All of my comments have been sufficiently addressed for this publication.

Response:

We thank the reviewer for the positive comments as well as the recognition of our effort in improving the manuscript.

Reviewer 2:

Comments to the Author:

Review queries are well answered and additional information is provided to support the answers.

Response:

We thank the reviewer for the positive comments as well as the recognition of our effort in improving the manuscript.

Reviewer 4:

Comments to the Author:

The authors have carefully revised their manuscript according to the reviewers' comments. The mechanism of how the steady growth state is achieved has been explained in a clearer way. I am pleased to recommend this paper for publication on Nat. Commun. in its current form.

Response:

We thank the reviewer for the positive comments as well as the recognition of our effort in improving the manuscript.